# Pilot Feasibility Assessment of a Tailored Physical Activity Prescription in Overweight and Obese People in a Public Hospital

**DOI:** 10.3390/ijerph191710774

**Published:** 2022-08-30

**Authors:** Janeth Tenorio-Mucha, Patricia Busta-Flores, Tania De la Cruz-Saldaña, Silvia Marcela Montufar-Crespo, German Malaga, Antonio Bernabe-Ortiz, Maria Lazo-Porras

**Affiliations:** 1CONEVID, Unidad de Conocimiento y Evidencia, Facultad de Medicina “Alberto Hurtado”, Universidad Peruana Cayetano Heredia, Lima 15102, Peru; 2CRONICAS Centre of Excellence in Chronic Diseases, Universidad Peruana Cayetano Heredia, Lima 15102, Peru; 3Knowledge and Evaluation Research Unit, Mayo Clinic, Rochester, MN 55902, USA; 4Division of Tropical and Humanitarian Medicine, Geneva University Hospitals, University of Geneva, 1205 Geneva, Switzerland

**Keywords:** obesity, overweight, exercise, feasibility studies, prescriptions

## Abstract

We aimed to evaluate the feasibility of a tailored physical activity (PA) prescription in overweight and obese people in a tertiary hospital in Lima, Peru. A feasibility pre–post-pilot study was conducted using mixed methods. Participants received a tailored prescription scheme for PA that lasted twelve weeks. It included two prescription sessions, three follow-up phone calls, and three evaluations. Primary feasibility outcomes were recruitment, visits, and phone call adherence. Primary intervention outcomes were self-reported PA levels and the 6 min walk test. Out of 228 people invited to participate, 30 were enrolled and received the first session of prescription, 11 went to the second session, and 21 went to the final evaluation; phone call participation decreased progressively during follow-up. There were no differences in the 6th week and the 12th week compared to the baseline for all the measures, except in the 6 min walk test. The participants considered the intervention was well designed, but they suggested complementing it with dietary instructions. The prescription of PA in overweight and obese people is feasible for promoting PA, but its implementation requires refinements to anticipate possible barriers to changing behavior.

## 1. Introduction

Physical inactivity is responsible for people being overweight, obese, and at least another 35 medical conditions [1]. It is also an important component in the increased morbidity and mortality of non-communicable diseases worldwide [2]. It is estimated that around one in three women and one in four men do not practice enough physical activity (PA) to stay healthy generating negative impacts on health systems, the environment, economic development, community well-being and quality of life [3]. 

Despite all the known benefits of PA on health and the recommendations of international guidelines [2,4,5,6], the rate of physical inactivity is high worldwide. A study including 51 countries of mainly low- and middle-income countries (LMIC) found that up to 52.6% of men and 72.0% of women are physically inactive [7]. The nutritional transition patterns show a preference for fast food, salty meals, and added caloric sweeteners. Rapid urbanization and the use of transportation to travel from one place to another instead of walking plus the appearance of technologies promoting physical inactivity have shifted the burden of obesity and non-communicable diseases (NCD) toward developing countries [8]. Additionally, personal barriers such as lack of motivation, limited time or resources, and fear of injury can hinder PA [9]. 

During daily clinical consultation, PA is provided as a brief or vague recommendation, dismissing its importance for people who are overweight, obese and for other non-communicable disease management. The prescription of PA in healthcare facilities and/or performed by healthcare staff is acceptable, feasible and cost-effective [10]. However, physicians and nurses consider the promotion of PA to be a time-consuming task and it is rarely seen as a priority during short-term clinical encounters, especially when there is a lack of official protocols and the perception of a lack of training or skills [11]. After compiling barriers and facilitators from patients and physicians, a review proposed a PA prescription process in three steps as follows, (i) perform an initial assessment of risk factors, readiness to change and exercise preference; (ii) develop a brief intervention including written instructions and a clear explanation on the frequency, intensity, type, timing and progression; and (iii) provide continued support in-person or via telephone involving adjustments and/or renewal of instructions, motivation and counseling [12]. 

Previous prescription interventions tested among physically inactive participants [13,14] and overweight older adults [15] demonstrated a positive effect on the quality of life and the reduction of body weight. These interventions encouraged the accumulation of the recommended 150 min of moderate-intensity PA per week. While promising, they were conducted in high-income countries in which PA prescription schemes have even been incorporated in their clinical guidelines. 

A survey in four Latin American countries suggests the need for tailored programs and interventions to increase PA considering the socio-economic characteristics of the population [16]. In Peru, barely 24.2% of the population performs moderate or vigorous PA, with women and the urban population being the most likely to be inactive [17]. Data from a tertiary hospital in Peru showed that one-third of consultations received counseling on PA, but instructions were brief and incomplete with no therapeutic goals [18]. Using recommendations previously reported in other studies as well as international guidelines, we design a prescription pilot intervention and evaluated its feasibility in overweight and obese people in a tertiary hospital in Lima, Peru. 

## 2. Materials and Methods

### 2.1. Study Design

A feasibility pre–post-pilot study was conducted using a mixed-method sequential explanatory design [19]. First, we collected and analyzed the quantitative data to provide information about PA uptake and accomplishment. Then, in a second strand, we conducted qualitative interviews to help explain the quantitative results obtained in the first strand. The rationale for this approach is that the quantitative data would provide a general understanding of the feasibility and the effects of the intervention while the qualitative data would explore in-depth experience-based information. This study was conducted between November 2018 and May 2019.

### 2.2. Participants

Using purposive sampling, we recruited participants from two outpatient clinics (family and internal medicine), and a diagnostic imaging center, all within a tertiary public hospital. This hospital is located in the north of Lima, the capital city of Peru. Currently, the hospital serves approximately 3 million people, mainly focusing on low-and middle-income populations [20]. 

Eligible participants were aged between 18 and 65 years, had a body mass index (BMI) ≥ 25 kg/m^2^ and <40 kg/m^2^, answered negatively about being physically active to the following previously validated question [21]: “In general, do you do moderate or intense exercise (such as hiking or sports) for at least 90 min per week?”, and expressed their willingness to perform physical activity. 

Participants were excluded if they self-reported a diagnosis or medical history of cardiovascular disease (type 2 diabetes mellitus and/or hypertension), had an ongoing pregnancy, were participating in a weight reduction program, or answered affirmatively to at least one question of the Physical Activity Readiness Questionnaire (PAR-Q+, 2018) [22], which evaluates the existence of health risks before starting a PA scheme. We excluded participants with cardiovascular disease history because they need specific exercise recommendations and exercise safety precautions that consider their medication, adverse effects during exercise, physical fitness as well as other health needs [23].

### 2.3. Intervention

The intervention was planned to last twelve weeks; a trained nutritionist specialized in PA and with previous experience working at gyms delivered the prescription. The prescription scheme was designed according to the World Health Organization (WHO) recommendations on physical activity: “Adults between 18 and 64 years should do at least 150 min of moderate aerobic physical activity per week or 75 min of vigorous aerobic activity or an equivalent combination of moderate vigorous activity” [2]. We organized a PA prescription scheme of 4 phases: warming-up, conditioning, cooling, and stretching. We drew up different options of exercises for each phase, offering suitable alternatives according to the Peruvian context. A minimum of 3 options to exercise a specific part of the body were proposed, which included the upper or lower body, legs, arms or joints. Frequency, duration, and intensity are detailed in Appendix A. We also prepared a printed guide with explanations of the prescription scheme which included pictures and detailed instructions to perform the exercises. A sports medicine professional validated the instructions of the guide. All participants were provided a color copy of this material and instructed to use it as a guide whenever they were scheduled to do PA. 

The prescription was delivered in two visits to provide a gradual scheme of exercises. The first included instructions for warming up plus aerobic exercises and the second added instructions for muscular strength, cooling, and stretching; each visit was planned to last between 40 to 60 min. If required, a demonstration of any of the exercises was performed by the nutritionist. At the end of each session, the participants were provided with a written prescription including the type of exercise, intensity, frequency and duration. Additionally, participants received general information regarding the benefits of PA on health.

### 2.4. Procedures

Before the intervention, we collected cardiovascular measures (heart rate and blood pressure), anthropometric measures (body weight, body mass index, waist circumference), the 6 min walk test, physical activity levels, and quality of life assessments. Then, using the information collected at baseline, the nutritionist and participants agreed on a tailored PA scheme, choosing the exercises from the printed material and considering each participant’s available space and time for exercise. 

Participants were followed-up by phone calls during the 1st, 3rd, and 9th weeks. Phone calls were carried out by the PA specialized nutritionist and a trained medical student, following a structured guide that contained motivational messages created by the research team. Each call lasted about 30 min: 10 min to strengthen the goal and prescribed scheme, 10 min of motivation for compliance to the scheme and to reminisce on the benefits of physical activity, and 10 min to inquire about the barriers and facilitators encountered by the participant. At the end of the 12th week, cardiovascular variables, anthropometric measures, the 6 min walk test, physical activity levels, and quality of life assessments were collected and we provided a last brief counseling on PA. Finally, we conducted in-depth interviews with six participants to explore their perceptions of the intervention. Figure 1 shows the flowchart of the procedures for this study. 

### 2.5. Assessment of Intervention Outcomes

All the assessments were conducted by trained fieldworkers and a standard operating procedure was developed to guide the steps in a standardized way. Our primary outcomes to assess the effect of the intervention were physical activity levels and the 6 min walking test. Physical activity was assessed using IPAQ-International Physical Activity Questionnaire Spanish version [24]. IPAQ provides information on sedentarism and physical activity of moderate and vigorous intensity. For baseline and final evaluations, we used the long version (27 questions) and for visit 2, a short version (7 questions). The items provided separate scores on walking, moderate-intensity and vigorous-intensity activities with values expressed in median MET-minutes/week. The computation of the scores was performed according to the Guidelines for data processing and analysis of the IPAQ [25]. The 6 min walk test [26] measures the maximum distance an individual can walk for 6 min walking as fast as possible. This test also evaluates the response of the respiratory system assessing the breath rate (BR), cardiovascular system assessing heart rate (HR) and blood pressure (BP) and the subjective perceived exertion using the Borg scale at rest, immediately after the exercise and after two and five minutes after exercise. We reported the walking distance, BR, HR, BP, dyspnea and fatigue plus the difference between the values immediately after exercise and at rest. 

Secondary outcomes were composed of anthropometric measures, cardiovascular variables and quality-of-life measurements. Anthropometric measures were weight (in kg), height (in m), body mass index (BMI), and abdominal circumference (in cm). Weight and height were measured with light clothes and no shoes using HBF-510LA Full Body Scale, and a wall-mounted stadiometer. BMI was estimated using the formula [weight (kg)/height (m^2^)]. Abdominal circumference was measured using a measuring tape around the waist at the level of the belly button. Cardiovascular variables were systolic (SBP) and diastolic (DBP) blood pressures (mmHg), measured according to American Heart Association (AHA) recommendations [27], and heart rate (beats/minute) measured with a pulse oximeter placed on the right index finger and recorded after 15 s of stabilized reading. Both were measured three times in intervals of 1, 2, and 5 min; we reported the average of the last two measurements. Finally, quality of life was assessed with the COOP/WONCA charts [28], which consist of 7 questions with illustrations and were administered by field workers. 

### 2.6. Assessment of Feasibility Outcomes

For this feasibility study, three outcomes were chosen: recruitment rate, adherence to visits, and adherence to phone calls. The recruitment rate was estimated based on the number of people who were approached and invited over the number of participants. Adherence to visits was calculated based on the number of participants who accomplished the evaluations in the 6th and 12th weeks. The adherence to phone calls was calculated based on the number of phone calls answered by participants over the twelve weeks of the intervention.

We collected qualitative data regarding the barriers and facilitators of the prescribed intervention by follow-up calls in the first, third and ninth weeks of intervention using an open-ended questionnaire. At the end of the intervention, we added in-depth interviews with six participants (~30% of final participants), choosing randomly, the three most and the three least adherent participants, all of them being women due to the high proportion of women in this study. We opted for this criterion to further explore and contrast participants’ views and experiences. A semi-structured interview was carried out by a psychologist with previous experience conducting interviews and focus groups in research about non-communicable diseases and mental health. The interviewer had no contact with the participants prior to the interview. The interview guide was developed by the coordination team (JTM, TDS, MLP), using a hybrid approach, we began with a deductively approach using topics reported in other studies on PA [29,30] and then inductively, we added other important topics that emerged in the quantitative data collection. No pilot to test the guide was performed, but it was reviewed by an experienced qualitative researcher to check its appropriateness. Topics included barriers and facilitators for adherence to prescription and follow-ups, beliefs about PA, strengths and challenges related to the scheme, and participants’ feedback about the intervention. On average, interviews lasted 30 min. 

### 2.7. Data Analysis

For quantitative analysis, all categorical variables were described in percentages, for numerical variables, means and standard deviation (SD), or median and interquartile range (IQR) were reported. To assess the differences of the PA prescription on the primary outcomes between baseline, 6th and 12th weeks, we used mixed linear regression models with random-effect intercepts using crude and sex- and age-adjusted models with robust standard errors and unstructured covariance to take into account any misspecification of dependence between measurements of the same individual. These models allow continuous and categorical variables and can model within-subject variability through random effects. Because of multiple comparisons, analysis was performed with a level of significance with Bonferroni correction for multiple comparisons, 0.025 for crude models and 0.017 for adjusted models. We used Stata 14 statistical package (StataCorp, College Station, TX, USA). 

For the qualitative analysis, the interviews were audio-recorded and transcribed verbatim. Deductive thematic analysis was used to assess the acceptability and feasibility of the PA prescription [31]. Data analysis was performed by one data coder. Checking data analysis was performed by a researcher with experience in qualitative methods. The transcriptions were read to identify coincidences, repetitions, similarities, or differences. Key excerpts were entered into a Microsoft Excel spreadsheet and presented in a narrative way. 

## 3. Results

### 3.1. Intervention Outcomes

At baseline, the mean body weight was 85.1 kg (SD: 12.9), the BMI mean was 33.4 kg/m^2^ (SD: 3.4), and the waist circumference mean was 104.2 cm (SD: 9.7). Among cardiovascular variables at baseline, average HR was 74.7 (SD: 9.5) beats per min, SBP mean was 108.1 mmHg (SD: 8.9), whereas DBP mean was 65.9 (SD: 8.2). Regarding PA according to the IPAQ; at baseline, the median of walking was 1386.0 (IQR: 2541.0) MET-min/week, whereas this estimate was 1305.0 (IQR: 2430.0) for PA of moderate intensity, and zero for PA of vigorous intensity. The median sitting time was 3 h/day (IQR: 3). All the COOP/WONCA scores of quality of life were under 4 points with the lowest scores in social activities. Details about primary and secondary outcomes are shown in Appendix A Appendix A. 

Comparing the measures across the intervention, there were no differences in the follow-up (6th week) and final evaluation (12th week) compared to the baseline regarding walking and moderate intensity activities, but there was a significant increase [Coef. 2646.3 (IC 95%: 691.9; 4600.8)] in vigorous intensity activities in the follow-up compared to the baseline. In the six-minute walking test, the HR difference immediately after the exercise and at rest were lower in the final evaluation compared with the baseline [Coef. −22.3 (IC 95%: −30.9; −13.7)]. The measures across the intervention in the crude model are shown in Appendix A. Among anthropometric measures, the waist circumference and DBP slightly increased in the final evaluation compared with the baseline but these findings were non-significant (Table 1).

### 3.2. Feasibility of the Intervention

We recruited 30 participants out of a total of 248 who were invited to participate, of this total, 197 were women and 71 were men, (recruitment rate: 12.1%), therefore 73.3% (22/30) were female, mean age was 38.6 (SD: 9.5), and 33.3% were currently employed. The attendance to the second session was 36.7% (11/30), the adherence to phone calls was 63.3% (19/30) for the 1st week, 50.0% (15/30) for the 3rd week, and 23.3% (7/30) for the 9th week. Finally, after an intense participant follow-up, which included home visits, we evaluated 70% (21/30) after week 12 (Figure 2). 

#### 3.2.1. Intervention Delivery Perceptions 

Interviewees expressed their satisfaction with the empathetic treatment given by the recruiters and the nutritionists during the entire program, but especially during the first visit. Each participant recognized they received clearly detailed exercises that were adapted to their daily routine; they liked that the instructions were complemented with demonstrations of the most difficult exercises. All respondents mentioned their satisfaction with the staff and the way the instructions were given. Nevertheless, patients claimed detailed diet counseling in addition to the exercise routine would help them improve their health. 

*“I am grateful for all the support they [Study staff]**have given me and what they have taught me about the guidelines, how to do everything with love. I told the nutritionist that I had to do it by myself and he showed me the guidelines with the booklet, how you are going to do it. And I did it just as he told me.”*—P16

Preferred exercises were aerobic and the least favored, stretching. Some participants disliked warm-up exercises because they considered them too time-consuming even though they were also considered easy. Pass-over exercises were muscular strength directed. Participants said that even though they discontinued the prescription, they tried to walk in their daily life. When asked about what they liked least about PA, some mentioned the sensation of heat when exercising and exercising during the summer. 

The participants stated that what they liked most were their structured physical activity routines, phone calls and visits. Most believed that more phone calls and visits would have helped them feel more accompanied and more motivated to do more PA. However, one of the participants mentioned that answering the follow-up calls made him feel annoyed as he did not like reporting he had not exercised or complied with the scheduled activities. 

*“For me it was perfect the amount and the time that elapsed between them. It was neither too heavy nor to the point of forgetfulness.”*—P46

All respondents expressed their satisfaction with the delivered material; they considered that the illustrations complemented the nutritionist’s instructions very well. The visual material was very helpful and some participants shared it with friends and family. Even so, the majority of the interviewees suggested that a video with the exercises would be more helpful. When asked for suggestions on how to improve the intervention, the participants requested they would like to receive a report of their evolution throughout the intervention and would also like to receive recommendations about diet and weight control. 

*“The manual was very didactic, and there was a personal who explained to me what I had to do and why I had to do it, and I also felt very well attended, very well attended by the professionals.”*—P46

#### 3.2.2. Barriers and Facilitators for PA 

The most important motivation to participate in the study was to lose weight and to reduce centimeters in waist circumference. Among the most adherent participants, the feeling of well-being drove them to complete the full intervention, although one of them mentioned that their main motivator to finish the intervention was the commitment assumed by the study team when agreeing to participate in the intervention. 

*“I was motivated by the fact that I want to lose weight, because I have vitiligo, and I have to take my food with me, that is, not to gain weight, but now I have gained weight, I am weighing 91.9, I was weighing 89.5 and I have gained weight (...) I have made a total imbalance of eating at odd hours, right? There... here… I was not at my eating time, that’s why I decided to participate”*—P44 

In general, participants felt comfortable doing exercises and being active. Once they started PA, participants said during the first days they experienced fatigue and muscular pain, which decreased gradually as the body adjusted to exercises. Additionally, some participants stated they felt ashamed when they were seen doing PA by their family. Most of the respondents preferred receiving family support to accomplish PA mainly from their children or partners but some expressed being on their own, living alone with no family support. Two participants mentioned feeling upset because their family mocked them when they exercised or made negative comments about their weight or diet. 

*“Yes, because the first few days I was shocked because I have always had an almost sedentary life and even more so with this disease I have. So, I could not have done nothing, I could have been in bed for almost more than a year... to start doing the exercises, that is, I realized that my body was not as my mind thought. I thought that I could still jump, that I could still do, for example, jump rope on both feet... and I realized that I could not because I am too overweight... I could not do it because I have too much weight.”*—P16 

According to the interviewees, the most important barrier for PA was time due to work or family duties. Other factors that hinder adherence to PA were fatigue, difficulty controlling their diet or anxiety about food and no family support. People who did not attend most visits and phone calls said it was because they did not have enough time to exercise. 

At the end of the study, participants mentioned that even though they did not lose weight, the PA helped them feel more active and good about themselves. Everyone expressed their desire to perform PA by participating in aerobic classes or sports. They asked to keep the material in order to continue using it. In general, they liked the feeling of well-being.

*“For example now I have knowledge of stretching exercises that I can do at home, the ailments have gone down as I tell you...I have a different rhythm, when I started walking I used to get very agitated, right? now I don’t get agitated the same way, my face doesn’t get hot anymore. So, I wanted to improve my physical condition, my mood and I have achieved it, right? (…) Well, I am enrolled in the gym, now I am working on project that needs some fieldwork, so I can keep doing my walks.”*—P46 

## 4. Discussion

The PA prescription planned for twelve weeks in participants with obesity or who were overweight had suboptimal adherence to follow-up visits and phone calls. This adherence decreased progressively. The prescription did not produce significant improvements in weight and blood pressure. However, participants considered the intervention was well designed with few recommendations for improvement. They strongly suggested complementing it with diet instructions. Among the participants, the main motivator for PA was weight loss and the main challenge in following the exercise program was a lack of time. 

The target population was a small proportion of the total individuals who were overweight or obese, which jeopardized the enrollment. Among the people excluded due to having pre-existing conditions; 76% (48/63) had hypertension or diabetes. Lifestyle interventions have benefits for people with obesity to prevent or reverse cardiometabolic abnormalities [32]. Therefore, some selection criteria should be relaxed to ensure appropriate recruitment because the physical activity may actually benefit overweight participants, particularly those with chronic metabolic diseases. 

Another challenge during recruitment was finding people with the willingness to participate in an intervention that involves PA. According to the Theory of Planned Behavior, intention is the main predictor of change in behavior [33], therefore, in the recruitment, we used the question: “Would you be willing to exercise regularly if told to do so?”. People, however, refused to participate in the study, stating a lack of time for exercise in the majority of the cases, and a lack of interest in the intervention to a lesser extent. Although intention is a predictor of PA behavior and we filtered participants motivated to exercise, our results showed how rapidly initial motivation wanes. 

Gollwitzer [34] argued that the adoption of positive behavior in health has a pre-intention stage in which a positive intention emerged as a motivator and a post-intention stage in which the intention is executed. The post-intention process needs action planning involving elements such as “when”, “how, and “where” plus strategies to anticipate risk situations or barriers that may hinder the execution of PA. In our pilot, the pre-intention stage comes from the participant’s motivation reinforced by messages about the benefits of PA from recruitment personnel and nutritionists. For the post-intention stage, the action planning was built jointly between nutritionists and participants with the intention to have greater chances of accomplishment. Nevertheless, we did not anticipate strategies to overcome the barriers that impede the execution of PA. The main personal level barriers identified in this pilot were lack of time due to work or family duties, fatigue, and difficulty controlling diet. Having identified them, the next step is to design ways to manage these difficulties and increase the chance of exercising. 

According to our results, most of the participants reported high levels of PA from the baseline to the final evaluation. In the pre-screening, we asked participants if they performed moderate or intense exercise for at least 90 min per week to filter them as physically inactive and being included in the pilot, therefore, we wrongly assumed that a low level of PA should characterize them. We used the IPAQ questionnaire to asses PA considering it is widely used and has shown good test and retest reliability [35,36]. We used the long version at baseline and final evaluation in order to collect the most accurate information, and the short version in the follow-up to reduce the time of evaluation for participants. The results show an increase in METs-min/week for vigorous intense activities in the follow-up compared to the baseline. A couple of systematic reviews point out that the IPAQ can overestimate the amount of PA [37] and it has a poor correlation with the total PA measured with accelerometers [38]. Thus, our results may be explained by those considerations plus the use of a short version of IPAQ in the follow-up and the long version in the baseline. It would be better to use the same version of the questionnaire throughout all the interventions. Additionally, during data collection, our field workers reported inconveniences phrasing the questions because some examples of PA included uncommon activities among the population of the study such as playing tennis, swimming, or yard work. Additionally, in some cases, participants found it difficult to distinguish between vigorous and moderate activities. 

We did not find improvements in weight, waist circumference, blood pressure, or quality of life outcomes; this could be because the change of PA was not significant throughout our intervention. However, the difference in the heart rate after and before the six-minute walk test was significantly smaller in the final evaluation than in the baseline evaluation. This cardiovascular finding may be explained by the autonomic adaptation to exercise [39], which is a phenomenon observed in athletes in whom endurance training decreases resting heart rate and cause a more rapid heart rate recovery following exercise. However, there is no conclusive evidence that training improves heart recovery after exercise and it may be influenced by other factors such as gender, age or exercise intensity [40]. Due to the lack of change in other cardiovascular variables, we cannot attribute that our intervention was responsible for the reduction in heart recovery in the six-minute walk test, although this is a positive signal of the incorporation of PA in daily life and a more accurate measure than self-report. 

Similar to our results, a study among women aged 40 to 74 [13] and the ProActive study [41] reported no significant change in weight, clinical or biochemical outcomes in comparison with the control group. However, both did show positive results in quality-of-life variables after 1 year of follow-up. Although our participants did not report improvement in their quality of life with the COOP/WONCA questionnaire, a few participants did mention that PA provided them with a feeling of well-being in the interviews. The results of the Swedish model of PA prescription concluded that changes in quality of life occurred independently of changes in PA, weight, or sedentary behavior [15]. 

Data from other interventions on weight management mention that the main motivator to recruit participants are novel aspects of the program and the way that it was presented, which also highlights that participants appreciate the friendly and non-judgmental attitude of healthcare staff [31]. Qualitative data revealed that participants were mostly satisfied with the components of the interventions, especially the first contact with healthcare staff, but this was not enough to ensure adherence to the PA prescription. Low motivation, lack of time, environmental, societal and social factors, health and physical limitations, negative thoughts, and socio-economic constraints are known barriers to lifestyle intervention adherence [30,42]. The most reported in our study was the lack of time due to work or family duties. 

In our study, the main motivation expressed by our participants was the wish to lose weight. Weight loss encourages people to continue practicing PA, eating healthy, and improving their quality of life [43]. It is reported that when people do not observe rapid changes in weight, they are less likely to be adherent [44]. This could be an important reason why our participants felt discouraged and stopped exercising. It is challenging to motivate people and engage them in exercise programs to pursue a long-term improvement —of their cardiovascular health—above short-term changes in weight. To reduce the high expectation on weight reduction it could be possible to emphasize on the prescription session that weight loss takes time and requires maintenance of PA behavior. 

Another aspect mentioned in the qualitative data is the importance of support from the microsystem level (family and friends). The lack of family support or even the bullying received by family members demoralizes participants to accomplish the prescription. A previous experience on patients with diabetes from a tertiary hospital in Peru found that participants feel that having supportive partners motivated them to do exercise and adopt good practices to manage their disease, but to reach this level of support, partners or family members need to be aware of the health condition of the patient [45]. In order to provide some kind of behavior, we included phone calls in the 1st, 3rd, and 9th weeks, which were well accepted by the majority of the participants. Even though this strategy helped to engage participants and reinforce the change of behavior, several providers of interventions raised concerns about the implementation of these actions in real clinical practice considering time and personnel constraints [31], which is a reality in public health centers in Peru. Hence, the intervention design researcher should not only consider elements of the intervention under research conditions but also in practice under real conditions. 

PA is important to reduce cardiovascular risk and mortality in the general population and in people who are overweight or obese [5]. Currently, PA is not prescribed as part of the usual care in the Peruvian health system and in other low- and middle-income countries [46]. We designed an intervention to prescribe the duration, type and frequency of different options of activities adapted to the context of each participant. Even though participant attrition was higher than we expected, we were able to identify the barriers they faced in order to perform their activities. Other barriers at mesosystem and ecosystem levels may also play a role: age, socio-economic status, type of job, social support and environment (places available and secure to do activities outside). Even when people understand the importance of exercise and how to do it, these factors can prevent PA. Multi-dimensional efforts are needed to increase exercise in the general population like interventions in schools, communities and the health system to decrease sedentarism. Addressing physical inactivity depends on several determinants that co-exist and interact among them; a comprehensive approach that may help to understand them is the socio-ecological model [47]. This study and others provide evidence that interventions need to be thought of at different levels (personal, interpersonal, micro and mesosystem, ecological). Therefore, assessing the casual determinants and interactions across these levels is needed to design a well-informed and contextually-appropriate combination of multi-level components in future interventions and public policies [48]. 

The major limitation of this study is the lack of a control group which may have allowed us to better determine any variation in the outcomes as a consequence of the intervention. We recognize the small sample of this study and their lack of representativeness, but this is a pilot and allows us to identify areas of improvement for future interventions. We also recognize the limitation of self-reported PA which overestimates the level of PA. Probably the use of devices such as accelerometers or pedometers may provide more accurate results, but they were not considered in this project due to financial constraints. On the qualitative exploration, the positive appreciation of the intervention could be overestimated due to desirability bias. 

We will explore and recommend incorporating a nutritional component to complement PA in future research. This will offer more benefits as well as enhance adherence to lifestyle modifications [30,43]. The evaluation of access to adequate places for PA and the objective assessment of psychosocial factors associated with PA are known to influence adherence to PA and need to be considered [42]. Additionally, implementing new practices such as exercise requires a change in behavior, therefore, to better guide the intervention development and evaluation process, it would be better to use some existing theory, framework or model. 

The main strength of our study is that we were able to deliver an individually tailored PA prescription according to the context of our participants even though the attrition rate was high. The collection of qualitative data gave us important feedback on the feasibility of a PA intervention in obese and overweight participants. We were able to explore the acceptability and main barriers of the prescription scheme. This information could be used to design other interventions to promote PA or the adoption of healthy lifestyles in similar contexts. 

## 5. Conclusions

Tailored prescriptions for physical activity among people with obesity and who are overweight are feasible, although, for scalability and sustainability they need to be complemented with diet instructions and more intensive follow-ups. To plan future interventions, it should be considered that despite the multiple benefits of physical activity, the main motivator for initiating exercise is weight loss and the main barrier to adherence to routines is a lack of time. Therefore, strategies should be designed to find or enhance other motivators to exercise and overcome time constraints.

## Figures and Tables

**Figure 1 ijerph-19-10774-f001:**
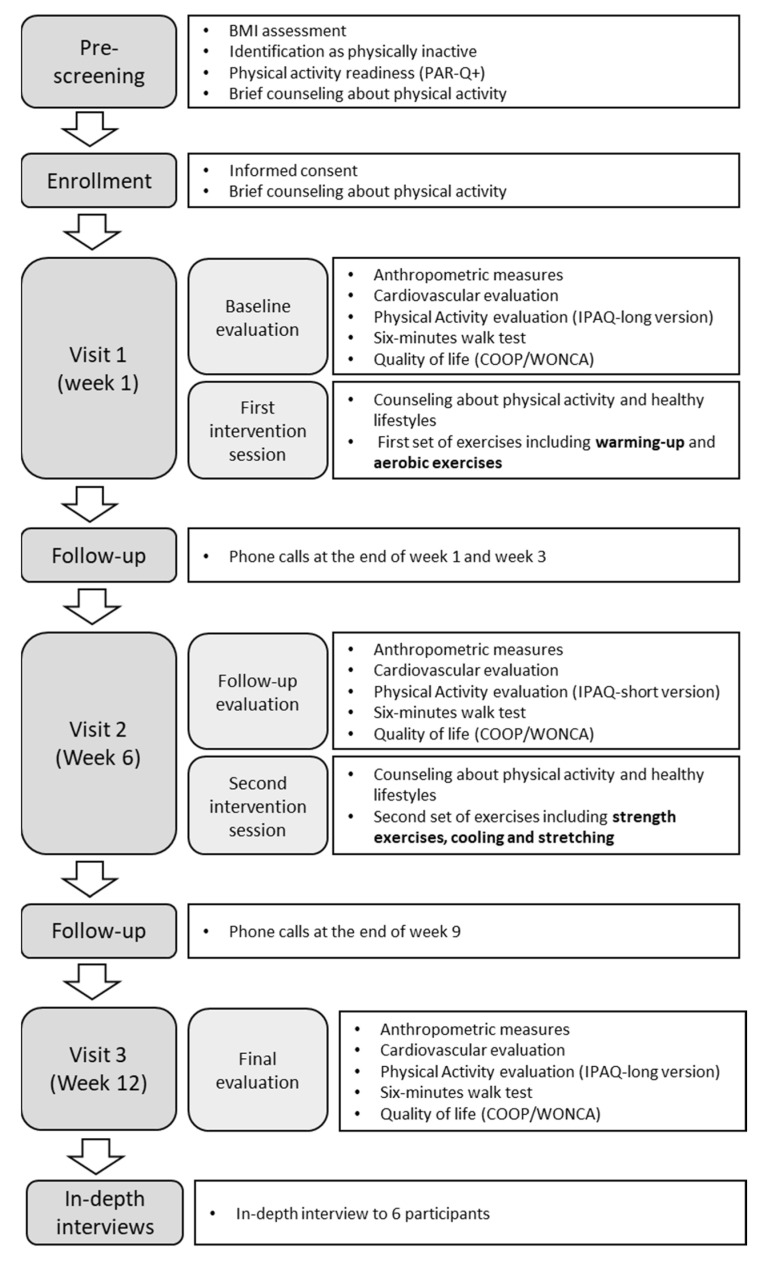
Physical activity prescription flow chart.

**Figure 2 ijerph-19-10774-f002:**
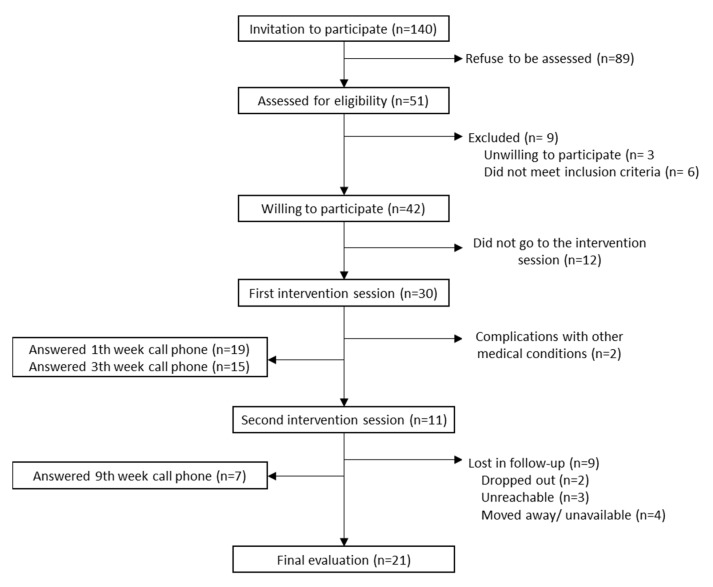
Recruitment and retention of participants.

**Table 1 ijerph-19-10774-t001:** Changes in primary and secondary clinical outcomes sixth and twelfth weeks compared with the baseline.

Outcome Measure	Model Adjusted by Sex and Age					
	Follow-Up (6th Week) N = 11			Final Evaluation (12th Week) N = 21		
	Coef. (IC 95%)		*p*-Value ^1^	Coef. (IC 95%)		*p*-Value ^1^
Primary						
Physical Activity (METs-min/week)						
Walking	−670	(−1778.4–438.5)	0.236	851.3	(−98.0–1800.7)	0.079
Moderate Intensity	−710	(−1588.0–166.7)	0.112	837.2	(−36.1–1710.4)	0.060
Vigorous Intensity	2646.3	(691.9–4600.8)	0.008	724.4	(−262.0–1710.7)	0.150
Sitting time (hours/day)	0.1	(−1.0–1.2)	0.847	0.1	(−1.4–1.6)	0.881
Six-minute walk test						
BR diff	−1.5	(−3.0–−0.1)	0.039	−1.1	(−3.0–0.9)	0.286
HR diff	−8.6	(−18.0–0.7)	0.071	−22.3	(−30.9–−13.7)	0.001
SBP diff	−3.2	(−9.1–2.7)	0.285	−0.4	(−6.9–6.2)	0.910
DBP diff	0.5	(−9.1–0.6)	0.867	−4.2	(−9.1–0.6)	0.087
Borg Scale diff						
Dyspnea	−0.7	(−2.1–0.6)	0.297	1.1	(−1.2–3.3)	0.346
Fatigue	−0.1	(−1.9–1.7)	0.918	1.3	(−0.4–2.9)	0.135
Walk Distance (m)	−12.6	(−51.0–25.8)	0.521	−45.4	(−86.9–−3.8)	0.032
Secondary						
Body weight (Kg)	0.3	(−0.5–1.1)	0.506	−0.2	(−1.4–1.0)	0.689
BMI (Kg/cm^2^)	0.2	(−0.2–0.5)	0.335	−0.1	(−0.6–0.4)	0.696
Waist circumference (cm)	−1	(−2.7–0.7)	0.248	−1.6	(−3.2–−0.02)	0.048
Heart rate	1	(−3.9–5.9)	0.685	0.6	(−3.1–4.3)	0.744
Blood pressure (mmHg)						
SBP	−0.2	(−4.3–4.0)	0.932	0.4	(−3.2–4.0)	0.843
DBP	−3.4	(−9.4–2.7)	0.279	3.6	(0.4–6.8)	0.027
Quality of Life						
Physical fitness	−0.2	(−0.7–0.2)	0.344	−0.04	(−0.5–0.4)	0.874
Feelings	0.6	(0.1–1.2)	0.033	0.3	(−0.3–0.9)	0.378
Daily activities	0.1	(−0.5–0.7)	0.709	0.4	(−0.2–0.9)	0.202
Social activities	0.6	(−0.1–1.3)	0.114	0.4	(−0.2–0.9)	0.212
Change in health	−0.5	(−1.0–0.1)	0.081	0.1	(−0.5–0.6)	0.858
Overall health	−0.3	(−0.8–0.3)	0.313	−0.3	(−0.8–0.2)	0.224

^1^*p*-value = 0.017 using Bonferroni correction. Note: DBP, diastolic blood pressure (mmHg); SBP, systolic blood pressure (mmHg); BR, breathing rate (breaths per minute), HR, heart rate (beats per minute).

## Data Availability

The data presented in this study are available on request from the corresponding author. The data are not publicly available due to confidentiality.

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
