# Peer review of "Pilot Feasibility Assessment of a Tailored Physical Activity Prescription in Overweight and Obese People in a Public Hospital"

_ijerph, 2022, doi:10.3390/ijerph191710774_

Round 1
Reviewer 1 Report
The effort made in this study has been intense but has not resulted in great contributions, mainly due to the small sample of the participating population. As the authors themselves say the selection criteria could be relaxed in this type of studies.
In the qualitative part, in-deep interviews, six subjects were selected, choosing the three most and least adherent participants. In order to find out the reason for the low participation it would have been advisable to interview the least adherent participants
Reviewer 2 Report
Congratulations to the authors for the theme under study. This report is well described and developed. However, some issues may need to be resolved. Some considerations are exposed.
The title is of vital importance for the selection and reading of research reports. It is recommended to identify the report as to the study design used, being coherent with the methodological procedures reported.
The introductory section presents an adequate contextualization of the object of study. However, a brief literature review of previous research on the topic would be useful. This would broaden the context and provide an empirical basis for the subsequent development of the hypotheses. In addition, we suggest explaining how this study aims to overcome the methodological limitations of previous studies, thus contributing to the justification and relevance of this study.
"promoting sedentary habits". Regarding the use of this term, it is suggested that the authors may distinguish the terms "sedentary behaviour" and "physical inactivity". It might make sense in the context of this research work.
Tremblay, M. S., Aubert, S., Barnes, J. D., Saunders, T. J., Carson, V., Latimer-Cheung, A. E., Chastin, S., Altenburg, T. M., Chinapaw, M., & SBRN Terminology Consensus Project Participants (2017). Sedentary Behavior Research Network (SBRN) - Terminology Consensus Project process and outcome. The international journal of behavioral nutrition and physical activity, 14(1), 75. https://doi.org/10.1186/s12966-017-0525-8
It is suggested to clarify which sampling method was used in the quantitative and qualitative phase.
The procedures described about qualitative data collection need to be improved. Was the interview script constructed inductively or deductively? Was a pilot study undertaken to check its appropriateness?
There is a high diversity of methods and techniques for qualitative data analysis. In addition, certain techniques are linked to certain epistemological positions. In this sense, it will be useful to inform readers, in more detail, about the procedures used in relation to qualitative data analysis. In addition, the use of procedures and techniques related to data credibility would also be useful (e.g. checking data quality across two, or more, researchers, using Cohen's Kappa coefficient inter-judge agreement).
In order to highlight the results, it is suggested that some quotes be inserted in the words of the participants, complementing the latent analysis carried out.
The Discussion is well organised and developed. Physical exercise and healthy eating are complex behaviours. Changes in these behaviours also depend on levels of influence and interaction involving individual aspects, perceptions of the environment and broader aspects such as public policies. To this extent, it would be interesting to theorize the results related to barriers to physical activity with other theoretical models, such as ecological models of physical activity.
